# Myocardial Inflammation, Sports Practice, and Sudden Cardiac Death: 2021 Update

**DOI:** 10.3390/medicina57030277

**Published:** 2021-03-17

**Authors:** Paolo Compagnucci, Giovanni Volpato, Umberto Falanga, Laura Cipolletta, Manuel Antonio Conti, Gino Grifoni, Giuseppe Ciliberti, Giulia Stronati, Marco Fogante, Marco Bergonti, Elena Sommariva, Federico Guerra, Andrea Giovagnoni, Antonio Dello Russo, Michela Casella

**Affiliations:** 1Cardiology and Arrhythmology Clinic, University Hospital “Ospedali Riuniti Umberto I-Lancisi-Salesi”, 60126 Ancona, Italy; giovol@live.it (G.V.); u.falanga@pm.univpm.it (U.F.); cipollettalaura@gmail.com (L.C.); clapson@hotmail.it (M.A.C.); ginogrifoni@gmail.com (G.G.); giuseppe.ciliberti@ospedaliriuniti.marche.it (G.C.); giulia.emily.stronati@gmail.com (G.S.); guerra.fede@gmail.com (F.G.); antonio.dellorusso@gmail.com (A.D.R.); michelacasella@hotmail.com (M.C.); 2Department of Biomedical Science and Public Health, Marche Polytechnic University, 60121 Ancona, Italy; 3Department of Radiology, University Hospital “Ospedali Riuniti Umberto I-Lancisi-Salesi”, 60121 Ancona, Italy; marco.fogante89@gmail.com (M.F.); a.giovagnoni@staff.univpm.it (A.G.); 4Department of Clinical Sciences and Community Health, University of Milan, 20138 Milan, Italy; bergmar21@gmail.com; 5Vascular Biology and Regenerative Medicine Unit, Centro Cardiologico Monzino IRCCS, 20138 Milan, Italy; esommariva@ccfm.it; 6Department of Clinical, Special and Dental Sciences, Marche Polytechnic University, 60121 Ancona, Italy

**Keywords:** myocardial inflammation, myocarditis, inflammatory cardiomyopathy, arrhythmias, ventricular tachycardia, sports, sudden cardiac death

## Abstract

Myocardial inflammation is an important cause of cardiovascular morbidity and sudden cardiac death in athletes. The relationship between sports practice and myocardial inflammation is complex, and recent data from studies concerning cardiac magnetic resonance imaging and endomyocardial biopsy have substantially added to our understanding of the challenges encountered in the comprehensive care of athletes with myocarditis or inflammatory cardiomyopathy (ICM). In this review, we provide an overview of the current knowledge on the epidemiology, pathophysiology, diagnosis, and treatment of myocarditis, ICM, and myopericarditis/perimyocarditis in athletes, with a special emphasis on arrhythmias, patient-tailored therapies, and sports eligibility issues.

## 1. Introduction

Inflammatory diseases of the myocardium encompass an important group of disorders characterized by heterogeneous clinical presentation, ranging from asymptomatic to sudden cardiac death (SCD), often occurring in otherwise healthy subjects [1]. The basic glossary includes myocarditis, defined as an inflammatory disease of the myocardium characterized by histological evidence of inflammatory infiltrates within the myocardium associated with myocyte degeneration and necrosis of non-ischemic origin, and inflammatory cardiomyopathy (ICM), defined as myocarditis associated with cardiac dysfunction [1]. When myocardial inflammation is accompanied by pericardial inflammation, the terms myopericarditis or perimyocarditis apply, the former indicating cases in which signs and/or symptoms of pericarditis prevail, whereas the latter are those cases characterized by prominent myocardial involvement, with evidence of systolic dysfunction [1].

Myocardial inflammation represents an important cause of morbidity and mortality among athletes and carries relevant implications for the assessment of sports eligibility. In recent years, we have witnessed a renewed interest in the complex intertwining relationships between sports practice and myocardial inflammation [2,3]. Furthermore, arrhythmic manifestations related to myocarditis have been studied in depth, with the publication of novel data on the characterization of such arrhythmias [4] and on the role of different therapeutic strategies, including immunosuppression and cardiac implantable electronic devices (CIEDs) [5].

In this review, we summarize the current knowledge on the epidemiology, pathophysiology, diagnosis, and treatment of inflammatory diseases of the myocardium from a sports cardiology perspective, according to published international recommendations [6,7]—when available—and the experience of a referral center for the care of athletes with myocardial inflammation.

## 2. Epidemiology

The precise ascertainment of the incidence and prevalence of myocarditis and ICM in athletes is complicated by the extreme variability of disease phenotype and by the fact that a diagnosis can only be achieved with advanced diagnostic tests, such as cardiac magnetic resonance (CMR) imaging and endomyocardial biopsy (EMB).

Autopsy series revealed a highly variable prevalence of myocarditis among sportsmen dying suddenly, ranging from none [8] to 42% [9]. A reasonable estimate of the contribution of myocarditis and ICM to the total burden of SCD in athletes ranges between 5% and 20%, [10,11,12]. Whether myocarditis is a more common cause of SCD in athletes than in sedentary age peers appears likely, but currently unproven [13]. The suboptimal sensitivity of the electrocardiogram (ECG) for a diagnosis of myocarditis and ICM may account for the substantial lack of high-quality data on myocarditis prevalence at the time of preparticipation screening. Recent studies shed light on the prevalence of silent myocardial/pericardial inflammation among athletes with mild or moderate coronavirus disease 2019 (COVID-19), revealing a low prevalence, ranging between 1% and 3% [14,15].

## 3. Pathophysiology

Similar to the relationship between various risk factors and cardiovascular events, physical activity and cardiovascular disease are linked by a J-shaped curve. Whereby moderate-intensity physical exercise exerts protective effects by improving many risk factors for atherosclerotic cardiovascular disease, repeated and prolonged bouts of intense exercise, typical of endurance sports practice (i.e., marathon running, cycling), may translate into a paradoxical increase in risk of both atherosclerotic and non-atherosclerotic cardiac events.

Several putative mechanisms may predispose athletes to develop myocardial inflammation. Intense sports practice in extreme climate conditions, for a prolonged duration, and in crowded spaces may impair protective immune responses, while at the same time facilitating the acquisition of upper respiratory tract [16] or gastrointestinal infections, which may even run completely asymptomatically [17]. Endurance sports, such as marathons, have been shown to impair both cellular (especially natural killer cells, neutrophils and macrophages) and humoral components (Immunoglobulin A [IgA] production) of the mucosal immune response to pathogens [17]. The acquisition of some specific pathogens may be facilitated by special sports activities, as in the case of Borrelia Burgdorferi and orienteering, or droplet-transmitted infections and boxing. Furthermore, close interpersonal contacts during major international sports events, or even competitions organized in endemic areas may boost the spread of infectious agents, as recently witnessed with SARS-CoV-2.

In some genetically predisposed hosts, the initial entry phase is followed by the systemic spread of pathogens, which may eventually infect the myocardium [18]. Myocardial injury ensues as a result of both direct viral cytotoxic effects (especially for adenovirus and coxsackie virus, which can directly induce myocyte lysis) [19], and immune-mediated mechanisms [18]. The relative contributions of direct and immune processes are difficult to disentangle in each patient, but immune mechanisms seem crucial in driving persistent inflammation and tissue repair with collagen deposition, which is the non-specific end-result of a wide range of different insults [20], and represents the mechanistic link to long-term complications (e.g., heart failure and arrhythmias) [21]. On the other hand, direct cardiomyocyte damage seems primarily implicated in the case of SARS-CoV-2, which can bind to the angiotensin converting enzyme 2 on cardiomyocytes and induce cell death [18].

Chemical triggers (especially cocaine and amphetamines) may be an underestimated and undeclared determinant of myocardial inflammation, given the widespread use of performance-enhancing drugs with the potential to induce myocardial injury, by both direct toxic and immune-mediated mechanisms [1,2,3].

The principal etiologies of myocarditis, ICM, and mixed pericardial–myocardial inflammatory syndromes are listed in Table 1 [1].

## 4. Clinical Signs and Symptoms

The clinical picture of myocardial inflammation is highly variable. Typical clinical scenarios include dyspnea, chest pain (especially in mixed pericardial–myocardial inflammation), fatigue, palpitations, dizziness, syncope, and sudden cardiac death [22]. An athlete-specific presentation of myocarditis or ICM may include a subtle deterioration of physical performance, which would not be otherwise noticed in a sedentary individual. On the other hand, a probably underestimated proportion of athletes may be completely asymptomatic, or even neglect symptoms out of fear of negative bearings on the competitive/professional sports career [3]. Non-specific prodromal symptoms (fever, gastrointestinal, and influenza-like symptoms) are commonly detected (in up to 80% of patients), and should be routinely asked for at the time of history taking.

In the practice of sports cardiology, a diagnosis of myocarditis or ICM often emerges in athletes after second-level diagnostic assessments are carried out for ventricular arrhythmias (premature ventricular complexes, couplets, nonsustained or sustained ventricular tachycardia) during ECG and/or exercise stress testing. In a recent study of active myocarditis, the prevalence of nonsustained ventricular tachycardia and sustained ventricular tachycardia/ventricular fibrillation were 28% and 7%, respectively [23]. The most common ECG morphology of myocardial inflammation-associated ventricular arrhythmias is right bundle branch block with superior axis (>50% of cases) [4,24], consistent with the common involvement of the inferior–posterior wall of the left ventricle. Ventricular arrhythmias have different features during the “hot” phase of myocardial inflammation as compared to prior (healed) myocarditis: ventricular fibrillation occurs almost exclusively, ventricular tachycardias are more commonly polymorphic and irregular, and premature ventricular complexes are more often polymorphic in acute than prior myocarditis [4,24], indicating that the electrical substrate is more unstable and complex during acute bouts of inflammation than in the chronic stage, when scar-related mechanisms prevail.

In our experience, most athletes with myocarditis present with a combination of ventricular and supraventricular arrhythmias at initial evaluation. Sinus tachycardia is common during the acute phase of myocarditis, due to the systemic inflammatory response and heart failure. Premature atrial complexes, potentially triggering atrial fibrillation or reentrant supraventricular tachycardias, and focal atrial tachycardias may reflect an underlying atrial myocarditis [25], which can even occur in the absence of ventricular inflammation [26,27]. As such, isolated atrial myocarditis could represent an underappreciated determinant of lone atrial fibrillation [28,29], and, in the chronic fibrotic phase, drive the conversion of paroxysmal to persistent/permanent forms of atrial fibrillation.

Atrio-ventricular block is another important presenting arrhythmia in patients with myocardial inflammation. First degree atrio-ventricular block is a common and nonspecific finding, being reported in up to 10% of patients with acute myocarditis [30], and in 7.5% of healthy athletes with a high resting vagal tone [31]. On the other hand, advanced atrio-ventricular block reflects a disrupted atrio-ventricular conduction, and is a rare but typical red-flag possibly associated to an underlying giant cell myocarditis, cardiac sarcoidosis, or bacterial (Lyme)/protozoal myocarditis, in which inflammatory cells may infiltrate and compress the conduction system. Therefore, second degree type II or third degree atrio-ventricular block should be carefully evaluated in otherwise healthy athletes [31].

The initial clinical presentation has been recently found to be associated with outcomes in a non-athlete cohort of young patients. More specifically, patients with left ventricular dysfunction (ejection fraction < 50%), sustained ventricular tachycardia, and hemodynamic instability at presentation have a markedly higher risk of death or heart transplantation at 5 years (14.7%) than subjects with none of these features (0%) [32].

## 5. Diagnosis

### 5.1. Endomyocardial Biopsy

Endomyocardial biopsy (EMB) remains the gold standard for a diagnosis of myocardial inflammation [1,33,34]. Not only does endomyocardial biopsy allow a definite diagnosis of myocarditis or ICM, while at the same time providing crucial information in the differential diagnosis with other cardiomyopathies [1,33,34,35], but EMB also informs prognosis and serves as a guide for therapy [1,33,34]. In fact, EMB enables the direct detection and quantification of viral genomes in the myocardium by real time polymerase chain reaction (PCR) techniques, thus distinguishing virus-positive versus virus-negative myocardial inflammation [1,33,34]. This information provides the foundation for a tailored treatment approach, as detailed in Section 6 [36]. Furthermore, the precise characterization of the inflammatory infiltrate (lymphocytic, eosinophilic, giant-cell, granulomatous (sarcoid)) is important for prognosis estimation: eosinophilic, giant-cell, and sarcoid myocarditis carry an ominous prognosis and usually require intense immunosuppression [36].

The classical Dallas criteria were only based on hematoxylin–eosin staining [1,33], and have been complemented by immunohistochemistry techniques (i.e., using antibodies for specific antigens, such as CD45 in leukocytes, CD68 in macrophages, CD3 and CD4 in T Helper cells, CD3 and CD8 in cytotoxic T cells, and CD19/20 in B cells) which have been found to increase diagnostic yield [1,36]. Furthermore, taking multiple samples (four to six) results in better sensitivity than taking fewer [1]. One important issue to be aware of with EMB is the risk of sampling error, whereby focal/patchy disease processes may not be adequately sampled, especially if specimens are blindly sampled from the interventricular septum, which is common practice. In case of structural/functional alterations confined to the left ventricle, this latter chamber should be sampled, using either a transseptal or a retrograde arterial (radial or femoral) approach [37]. Furthermore, electroanatomical voltage mapping (EVM) can be performed during the same procedure as EMB [38], and has been found to further enhance EMB’s diagnostic yield [39,40], by allowing direct sampling of myocardial segments showing abnormal voltages. EVM-guided EMB currently represents the preferred technique at our institution.

Being an invasive test, EMB carries some risks; however, in experienced hands, EMB is safe, and complications are mainly limited to vascular access-related issues [40,41]. 

According to a 2013 position statement issued by the European Society of Cardiology (ESC) [1], EMB should be considered in each case of clinically suspected myocarditis, and should be repeated in order to assess the response to specific therapeutic measures or in patients with a suspected false negative initial result [1]. Recent recommendations provide more judicious indications [36], suggesting limiting EMB to the following clinical settings:(1)patients with suspected acute myocarditis and severe heart failure or cardiogenic shock;(2)suspected acute myocarditis complicated by severe myocardial dysfunction, acute heart failure (HF), ventricular arrhythmias, or high-degree atrioventricular block;(3)suspected acute myocarditis or suspected chronic inflammatory cardiomyopathy associated with peripheral eosinophilia;(4)suspected acute myocarditis or chronic inflammatory cardiomyopathy with the persistent or relapsing release of biomarkers of myocardial necrosis, particularly if associated to a suspected/known autoimmune disorder or ventricular arrhythmias or high-degree atrioventricular block; and(5)myocarditis in the setting of immune checkpoint inhibitors, where appropriate diagnosis has implications for patients receiving additional cancer therapy.

When dealing with athletes, in the absence of specific data from large series, we strongly believe that EMB (ideally, EVM-guided EMB) should always be performed in experienced centers in the case of clinically suspected myocarditis, especially in the case of competitive or professional athletes, in whom information derived from EMB may prove crucial in terms of therapeutic decisions and sports eligibility assessment [42,43].

Unfortunately, EMB is still an underused diagnostic test, and the lack of a widespread availability of EMB represents an important impediment to a modern, tailored approach to the treatment of myocarditis and inflammatory cardiomyopathy.

### 5.2. Electrocardiography

12-lead ECG is a low-cost, widely available test, and it is a key component of pre-participation screening in Italy and other countries [7], and ECG abnormalities represent one of the ESC diagnostic criteria for clinically suspected myocarditis (Table 2) [1]. Non-athlete patients with myocarditis or ICM present with abnormal ECGs in >85% of cases [36]. Nonetheless, ECG findings in patients with ongoing myocardial inflammation are quite nonspecific, and include ST segment elevation or, less commonly, depression, as well as arrhythmias, as detailed in Section 4 [1,4,24]. Patients with perimyocarditis or myopericarditis usually display a typical temporal pattern of ECG evolution, initially characterized by diffuse ST segment elevation and PR depression, followed by ST segment normalization, and finally diffuse T wave inversion, which reflects inflammation of the epicardium [44].

Careful interpretation of the ECG in athletes relies on knowledge of training-associated benign variants and their distinction from abnormal patterns [31]. For instance, an elevation of the J point by 0.1 mV or more, with or without a slurring or notch of the terminal portion of the QRS in lateral and/or inferior leads is a common and benign finding in sportsmen and is referred to as early repolarization [31]. This pattern is even more common in athletes of Afro-American descent, and should be regarded as a normal variant, with no need to obtain further imaging/electrophysiological tests [31]. Ambulatory ECG monitoring for 24 (or, preferably 48) hours, external event monitors, or implantable loop recorders may allow a better assessment of arrhythmia burden.

### 5.3. Laboratory Tests

Myocardial injury, as defined by increased high-sensitivity cardiac troponin levels, is common in patients with acute myocarditis, but the degree of troponin elevation correlates poorly to the severity of myocardial dysfunction [36], and myocardial injury may be absent or subtle in chronic myocarditis. Nonetheless, increased troponin levels are one of the ESC diagnostic criteria for a diagnosis of clinically suspected myocarditis (Table 2) [1]. Furthermore, a diagnosis of perimyocarditis or myopericarditis relies on the detection of increased troponin levels, in addition to the ESC criteria for pericarditis (two or more among chest pain, pericardial rubs, ECG changes, and new pericardial effusion) [1,44]. The assessment of myocardial injury in athletes is complicated by the fact that troponin release has been well documented after intense training/competitions.

PCR and erythrocyte sedimentation rate are usually increased in the acute phase, reflecting systemic inflammation [36]. The complete blood count may reveal eosinophilia or lymphophenia, pointing to eosinophilic myocarditis or systemic lupus erythematosus (SLE), respectively. Antinuclear antibodies may be useful to confirm a diagnosis of SLE.

The utility of viral serology has been challenged in the ESC position statement [1], due to the scarce correlation between peripheral antibody titers and PCR studies on EMB samples, HIV and Borrelia Burgdorferi antibodies being an exception. A positive serology for cardiotropic viruses may be particularly common among athletes due to their unique lifestyle (i.e., frequent travels and contacts with other individuals during competitions).

Finally, toxicological screening may prove useful to detect performance-enhancing drugs (i.e., amphetamines or cocaine), which may represent an undeclared trigger of myocardial inflammation.

### 5.4. Echocardiography

Functional or structural abnormalities are common in patients with myocarditis, and constitute another ESC diagnostic criterion for clinically suspected myocarditis (Table 2) [1]; therefore, transthoracic echocardiography is an important component of diagnostic workup. The echocardiographic picture is heterogeneous, with dilated, hypertrophic or restrictive phenotypes reported [1,36]. Even when global systolic function is normal, altered echogenic texture, regional inferior–lateral hypokinesis, parietal hypertrophy, or a small pericardial effusion may point to a diagnosis of myocarditis [36]. A small, hypertrophic, hypokinetic left ventricle is typically encountered in fulminant myocarditis [1,36], whereas chronic inflammatory cardiomyopathy is characterized by a dilated left ventricle with impaired systolic function and normal or moderately increased wall thickness [36]. Myopericarditis and perimyocarditis are both characterized by the presence of pericardial effusion and brightness, and can be differentiated according to the absence or presence of regional/global left ventricular (LV) dysfunction, respectively [44].

For a correct interpretation of echocardiography in athletes, physicians should understand the profound impact of regular sports practice on cardiac structure. Isotonic sports impose a significant volume load on the heart, which adapts through chamber dilation (left ventricular end diastolic diameter up to >60 mm in élite cyclists) and/or mild eccentric hypertrophy (septal thickness up to 12 mm) [2,45], which may be difficult to distinguish from an inflammatory cardiomyopathy [2,45]. Noninvasive tissue characterization (with cardiac magnetic resonance (CMR) imaging), EMB, the assessment of the response of wall motion abnormalities to exercise, and an evaluation of the proportionality between cardiac dilation and exercise capacity may be necessary for a precise differential diagnosis [2,45].

### 5.5. Cardiac Magnetic Resonance

CMR has emerged as the preferred noninvasive diagnostic test for the assessment of cardiac structure, function, and tissue characterization in athletes with a suspected myocarditis or ICM [1,2,3,36]. Accordingly, CMR may provide information to fulfill two of the four ESC criteria for a diagnosis of clinically suspected myocarditis (Table 2) [1], by providing both gold standard morpho-functional information, and allowing the noninvasive detection of oedema and fibrosis [46].

Updated CMR requirements for a diagnosis of myocarditis or ICM include both T2-based criteria (T2-mapping or T2 weighed images), which reflect myocardial oedema, and T1-based criteria (T1-mapping, extracellular volume, and late gadolinium enhancement (LGE)), which mainly detect necrosis and fibrosis (Figure 1) [46]. The use of these updated combined criteria allows high specificity (>94% in most published reports) and sensitivity (>90% in most published reports) [46]. The presence of pericardial effusion, pericardial hyperintensity in LGE, T1, or T2 mapping, and regional or global wall motion abnormalities represent ancillary supportive findings [46]. Pericardial effusion *per se* may reflect the presence of heart failure and is not sufficient to support a diagnosis of myopericarditis/perimyocarditis, which are characterized by signs of pericardial inflammation (pericardial thickening and hyperintensity in LGE, T1, or T2 mapping).

The assessment of athletes with isolated LGE areas is a clinical conundrum: a CMR study of marathon runners has shown a 7% prevalence of non-ischemic LGE [47]. Whether LGE is the expression of an inflammatory cardiomyopathy, a mere consequence of marathon-induced myocardial injury, or a combination of the two remains speculative, as well as the prognostic weight of LGE [2,45,46,47]. The pattern of LGE distribution provides important prognostic information: inferior–lateral subepicardial stria has been reported to be associated with a high risk of malignant ventricular arrhythmias in athletes (Figure 1) [48], whereas septal mid-layer LGE has been found to be predictive of adverse events in non-athletes with myocarditis [2]. However, isolated LGE can be found in a plethora of non-ischemic cardiomyopathies, including arrhythmogenic right or left ventricular cardiomyopathy, and EVM-guided EMB has a powerful role in the differential diagnosis [49].

The threshold for CMR should be low in athletes, and CMR should be performed when the clinical picture is potentially consistent with a myocarditis or ICM, in case of ventricular arrhythmias, or when other non-invasive tests show unclear findings. A possible exception could be represented by critically ill patients, in whom CMR might not be logistically feasible; in these cases, EMB provides crucial information for clinical management [36]. After a diagnosis of myocarditis, CMR should be repeated (after 6–12 months) to assess the response to treatment [36].

### 5.6. Other Diagnostic Tests

Athletes presenting with an acute picture, characterized by symptoms, ECG changes, and/or imaging abnormalities, should undergo invasive coronary angiography, in order to rule out obstructive coronary artery disease [50]. Only then a diagnosis of myocarditis can be safely considered, further assessed with CMR, and confirmed by EMB. A coronary computed tomography angiography may be considered an adequate and less invasive alternative for young athletes without risk factors for coronary artery disease [7].

Nuclear imaging studies (mainly positron emission tomography, PET) have an ancillary role, and should be obtained in case of a suspected cardiac sarcoidosis [36,51].

### 5.7. Differential Diagnosis

The differential diagnosis of myocardial inflammatory diseases in athletes is wide and includes the athlete’s heart, dilated, arrhythmogenic or hypertrophic cardiomyopathies [1,6,7], and ischemic heart disease (usually due to anomalous origin of a coronary artery or myocardial bridge in younger subjects, and to atherosclerotic coronary artery disease in senior athletes) [1,6,7,35]. CMR, coronary angiography, and genetic testing may be required, as well as a clinical-imaging reassessment after a 3–6 month period of deconditioning [7]. Furthermore, among athletes presenting with ventricular arrhythmias, idiopathic benign outflow tract or fascicular arrhythmias are common and should be distinguished from ventricular arrhythmias in the context of structural heart disease on the basis of CMR and EVM findings [7,42,43].

## 6. Comprehensive Patient Care

### 6.1. Medical Management

Information derived from EMB lays the foundation of patient-specific management [1,49]. Although cause-specific treatments have not yet been tested in rigorous, randomized, controlled trials, accumulating evidence is supporting a tailored EMB-based approach, according to the presence, type, and load of viruses in EMB samples, and to the type of inflammatory infiltrates [36,52]. In the absence of specific data, medical management is similar for athletes and non-athletes with myocarditis or ICM.

Virus-negative chronic ICM should be treated with a combination of prednisone and azathioprine [36,51], which has been shown to be associated with myocardial recovery in several studies. In our experience, prednisone and azathioprine are well tolerated in competitive and elite athletes. Although such a combination appears reasonable even in patients with virus-negative acute myocarditis and should be used in patients with associated systemic autoimmune disorders (in whom intravenous immunoglobulin, cyclophosphamide, or rituximab may replace azathioprine due to their more rapid onset of action), there is a paucity of data in support of immunosuppression in patients with isolated lymphocytic acute myocarditis [36,52]. Corticosteroid-based immunosuppression is a well-established treatment option for patients with eosinophilic or giant cell myocarditis, or cardiac sarcoidosis [52].

In regard to virus-positive chronic ICM, there is limited evidence in support of interferon beta for patients with EMB evidence of adenovirus or enterovirus [52]. Parvovirus B19-associated ICM has been treated favorably with intravenous immunoglobulins, telbivudine, or even prednisone plus azathioprine regimens in the case of low viral load and absence of virus replication [52,53]. Some experts recommend a combination of immunosuppression and antiviral agents (ganciclovir, acyclovir, or valacyclovir) for selected patients with human herpesvirus 6-associated acute myocarditis [52], whereas no data in support of specific therapies for other virus-positive acute myocarditis are currently available.

Standard heart failure regimens (beta blockers and angiotensin converting enzyme inhibitors/angiotensin receptor blockers) should be prescribed to every patient with chronic ICM or hemodynamically stable acute myocarditis with evidence of systolic dysfunction [1,36]. However, the use of beta blockers may result in worse exercise performance and increased perception of fatigue in athletes practicing endurance sports [54], while beta blockers are prohibited by the World Anti-Doping Agency for some skill sports, including golf, archery, and shooting, due to their relieving effect on tremor.

The management of patients with myopericarditis involves the administration of aspirin (1–1.5 g/day) or nonsteroidal anti-inflammatory drugs (ibuprofen 1.2 g/day) in order to allow symptom control [44]. The use of these drugs is discouraged in isolated myocardial inflammation, based on evidence of a worse outcome in experimental murine models [44]. The use of colchicine, a widely used agent in patients with pericarditis, is not currently supported by strong scientific evidence in the setting of myopericarditis/perimyocarditis; nonetheless, colchicine may help reduce the risk of recurrences [1,44].

### 6.2. Cardiac Implantable Electronic Devices, Catheter Ablation, Mechanical Circulatory Support, and Heart Transplantaion

The indication for the implantation of implantable cardioverter defibrillators (ICDs) in patients with acute myocarditis depends on several factors, which include the presence of ventricular arrhythmias, presence and localization (septal) of LGE by CMR [55], the extension of low-voltage areas by EVM (especially if >10%) [5], and EMB evidence of giant cell myocarditis or cardiac sarcoidosis [36]. ICDs are usually implanted prior to discharge in patients presenting with sustained ventricular arrhythmias (VAs) (secondary prevention), whereas the timing of implantation is more variable in the case of primary prevention. In patients with chronic ICM, the indications are similar to those for heart failure with reduced ejection fraction [36]. Patients with second degree type 2 atrioventricular block or third degree atrioventricular block should receive a dual chamber pacemaker; patients with cardiac sarcoidosis and an indication for pacing should receive a dual chamber ICD, due to the recognized risk of sudden death [56]. ICD programming in athletes should include high-rate cutoffs and long-detection duration, in order to reduce inappropriate shocks during physical exercise [57].

Catheter ablation may be considered among patients with recurrent ventricular tachycardia; however, according to recent data, the risk of recurrences is highest if catheter ablation is performed during the acute phase of myocarditis and might preferably be performed after the healing of acute inflammation [58].

Short-term mechanical circulatory support devices (intra-aortic balloon pump, Impella, or veno-arterial extracorporeal membrane oxygenation) are commonly used in the case of refractory heart failure or cardiogenic shock, given the high likelihood of recovery among patients with fulminant myocarditis [36].

Long-term mechanical circulatory support or heart transplantation should be considered when patients with acute myocarditis cannot be weaned off short-term mechanical circulatory support devices [36], or when patients with chronic ICM transition to advanced heart failure.

### 6.3. Sports Eligibility Assessment

Athletes with myocarditis should refrain for participation in competitive or leisure time sports for 3–6 months, a time window during which the practice of sports may augment risk of sudden cardiac death and worsen myocardial inflammation [1,6,7]. Similar recommendations apply to the management of athletes with myopericarditis/perimyocarditis [7,44]. There is lack of information on the impact of low–moderate intensity training and isometric exercise during the acute phase, and a case-by-case evaluation should be performed [2]. After the 3–6 month time window, athletes should be re-assessed with laboratory exams (including troponin and inflammation markers), exercise stress testing (in case of normal troponin levels), ambulatory ECG monitoring, and CMR; athletes may be deemed eligible for competitive or leisure time sports in the case of the absence of complex ventricular arrhythmias (>2000/day or exercise induced premature ventricular complexes, couplets, nonsustained or sustained VT), normal ejection fraction, and the absence of LGE [2,6,7]. Among athletes with persistent LGE at follow-up CMR scans, sports practice may be considered in the case of the limited extension of LGE (<20%) [7].

Due to the potential risk of cardiovascular sequelae [59] which are sometimes related to myocardial inflammation, expert consensus recommendations for the management of competitive athletes after COVID-19 have been recently provided [60]. According to these recommendations, athletes with asymptomatic or mildly symptomatic SARS-CoV-2 infection should gradually resume exercising after 10 days of complete abstinence, whereas among athletes with prior moderate–severe COVID-19, a full cardiovascular evaluation (including at least clinical assessment, ECG, echocardiography, and troponin levels) is advocated prior to resuming training [61]. These suggestions might well be extended to other viral infections potentially associated to myocardial inflammation.

In the absence of specific data, athletes with chronic ICM should be managed similarly to other patients with dilated cardiomyopathy, and may be considered eligible for competitive or leisure time high-intensity sports in the case of the absence of limiting symptoms, lack of complex arrhythmias on ambulatory ECG monitoring and exercise stress testing, an ejection fraction >45%, and absent or limited (<20%) LGE [7].

The practice of sports among patients with an ICD implanted after a myocarditis or in case of inflammatory cardiomyopathy might be allowed on a case-by-case basis, taking into account the underlying myocardial substrate (extension of LGE and low voltage area by EVM), avoiding direct impact on the implanted device (by padding and abstaining from contact sports), and averting sports which entail dangerous circumstances (climbing, motor sports, or driving) [6,7].

After cardiac transplantation, there are some isolated reports suggesting that participation in low–moderate intensity leisure time or competitive sports is feasible, whereas endurance sports should be avoided, given the lack of chronotropic competence due to graft denervation [7].

## 7. Conclusions

Myocardial/pericardial inflammation is relatively common in athletes, and can portend potentially ominous complications, including sudden cardiac death. A thorough patient evaluation, including CMR, and (EVM-guided) EMB is key for successful management and can provide essential data for the safe assessment of sports eligibility and for the estimation of prognosis. The care athletes with myocarditis or chronic ICM is best accomplished in centers with considerable expertise [60], given the multiple challenges encountered in the diagnosis, treatment, and follow-up of these patients.

## Figures and Tables

**Figure 1 medicina-57-00277-f001:**
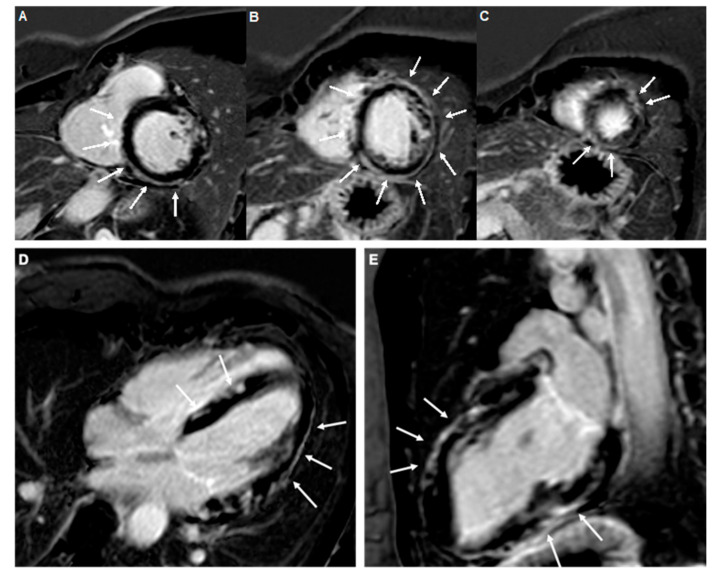
A prototypical example of extensive myocardial scarring due to myocarditis in a 35-year-old female tennis player. Contrast-enhanced cardiac magnetic resonance images (**A**–**C**), short axis views, (**D**), four-chamber view; (**E**), two-chamber view, showing an almost circumferential subepicardial stria of late gadolinium enhancement involving the interventricular septum, inferolateral, lateral, and anterolateral walls of the left ventricle (white arrows).

**Table 1 medicina-57-00277-t001:** Principal etiologies of myocarditis, inflammatory cardiomyopathy, perimyocarditis, and myopericarditis.

**Viruses**
Parvovirus B19 Herpesvirus (Human Herpesvirus 6, Cytomegalovirus, Epstein–Barr virus) SARS-CoV-2 Adenovirus Enterovirus (coxsackie, echovirus) Influenza Human Immunodeficiency virus Hepatitis C virus
**Bacteria**
Borrelia Burgdorferi Staphilococcus Streptococcus Trypanosoma Coxiella Burnetii
**Other Infectious Agents**
Echinococcus Taenia Toxocara Trichinella Trypanosoma cruzi Aspergillus
**Toxic Agents**
Amphetamines Cocaine Insect stings Spider/snake bites
**Immune Mechanisms**
Associated with systemic autoimmune disorders (e.g., systemic lupus erythematosus, vasculitides, sarcoidosis)
Allergen mediated (vaccines, drugs, serum sickness)

**Table 2 medicina-57-00277-t002:** Definition of clinically suspected myocarditis according to the European Society of Cardiology.

**Suggestive Clinical Presentation, Defined as at Least One of the Following:**
Acute chest pain syndrome Cardiogenic shock New-onset heart failure (<3 months) Chronic heart failure (>3 months)
**Diagnostic Criteria, Defined as at Least One of the Following:**
New QRS, ST-T abnormalities, new arrhythmias (both brady- and tachy-) Myocardial injury (increased troponin levels) Morpho-functional abnormalities (by echocardiography of cardiac magnetic resonance imaging) Noninvasive tissue characterization criteria by cardiac magnetic resonance imaging (T2-based criteria plus T1-based criteria)

Clinically suspected myocarditis is defined by at least one suggestive clinical presentation plus at least one diagnostic criterion.

## Data Availability

Data is contained within the article.

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
