# Peer review of "Myocardial Inflammation, Sports Practice, and Sudden Cardiac Death: 2021 Update"

_medicina, 2021, doi:10.3390/medicina57030277_

Round 1

Reviewer 1 Report

The review manuscript by Compagnucci et al,  is well written and informative.

Some minor comments:

  1. Table one: Those are the principal causes of myocarditis in general, not exclusive of athletes as mentioned in the title.
  2. Point 5.2: The baseline electrocardiogram in athletes is usually not normal even without myocarditis. ST elevation is a common finding in athletes. In a review manuscript it is expected to provide clear information for clinical decisions. The current paragraph is short and not really informative, no examples are given.
  3. Please add a section of differential diagnosis

Author Response

Response to Reviewer 1 Comments

Point 1: Table one: Those are the principal causes of myocarditis in general, not exclusive of athletes as mentioned in the title.

Response 1: We thank the respected reviewer for his/her comment. As of today, we do not have high-quality data on the etiology of myocardial inflammation in athletes as compared to the non-athletic population. Accordingly, we have modified the table’s title by removing “in athletes”, while adding “perimyocarditis, and myopericarditis” (page 3, lines 120-121). In fact, mixed pericardiac-myocardial inflammatory syndromes recognize similar etiologic agents to myocarditis/inflammatory cardiomyopathy (Imazio M, Cooper LT. Management of myopericarditis. Expert Rev Cardiovasc Ther 2013, 11(2):193-201. doi: 10.1586/erc.12.184. PMID: 23405840).

Point 2: Point 5.2: The baseline electrocardiogram in athletes is usually not normal even without myocarditis. ST elevation is a common finding in athletes. In a review manuscript it is expected to provide clear information for clinical decisions. The current paragraph is short and not really informative, no examples are given.

Response 2: Thank you for your insightful comment. In the revised version of our manuscript, we have added more information on ST segment elevation in athletes, by describing both myopericarditis/perimyocarditis (page 6, lines 313-317) and the early repolarization pattern (page 6, lines 319-324). Furthermore, we provided reference to the international recommendations for electrocardiographic interpretation in athletes (Sharma S, et al. International Recommendations for Electrocardiographic Interpretation in Athletes. J Am Coll Cardiol 2017, 69(8):1057-1075), in which common benign variants encountered in athletes are extensively described (page 6, lines 318-319).

Point 3: Please add a section of differential diagnosis

Response 3: Thank you for your kind request. We have now added a paragraph (5.7) concerning the differential diagnosis of athletes with myocarditis/inflammatory cardiomyopathy (page 9, lines 455-465).

Reviewer 2 Report

The manuscript is quite well written on the topic very interesting for sport medicine doctors and cardiologists. In general, the authors summarizing our not new knowledge on myocarditis and inflammatory cardiomyopathy at the same time with only a minor focus on athletes. 

I have some major remarks:

  1. As we have similar articles on myocarditis in the general population thus specific focus on athletes (it is not clear which athletes’ manuscript is addressing – competitive/recreational/endurance etc.) would improve the manuscript largely. Now it seems that article was written originally about acute myocarditis in general population and then has been updated a little with some special aspects in unspecified athletes.
  2. The manuscript is too long and the Special Issue "Sudden Cardiac Death in Athletes" topic is not comprehensively addressed. The manuscript should be more focused on athletes and sudden cardiac death in the setting of myocarditis/perimyocarditis/inflammatory CMP.
  3. As we often face perimyocarditis I the acute face of diseases this should be mentioned and discussed.
  4. More information on COVID-19 myocarditis in athletes is needed taking into account current pandemics.

Minor remarks:

  1. Please discuss on specific etiological factors in special sporting activities – orienteering (exposure to borreliosis), boxing (droplet infection), a lot of contacts during global sporting events, sporting activities in endemic areas and etc.
  2. Please discuss sensitivity and specificity of CMR to detect myocarditis.
  3. Please discuss NSAIDs in the setting of myocarditis; BB in athletes.
  4. Please stress that EMB should be performed in experienced centers…what is complication rate in athletes of EMB?
  5. Please include coronary artery CT angiography as a imaging test that could replace invasive angiography in athletes.
  6. Please discuss more the timing of device implantation in acute myocarditis.
  7. Please talk about prevention measures in athletes (training activities during viral infections and etc., evidence of ICD shocks in athletes and etc.)

Author Response

Response to Reviewer 2 Comments

Major Remarks:

Point 1: As we have similar articles on myocarditis in the general population thus specific focus on athletes (it is not clear which athletes’ manuscript is addressing – competitive/recreational/endurance etc.) would improve the manuscript largely. Now it seems that article was written originally about acute myocarditis in general population and then has been updated a little with some special aspects in unspecified athletes.

Response 1: We wish to thank the respected reviewer for his/her comment. However, our article was planned and written to provide a general overview of the variable and heterogeneous clinical presentation of inflammatory diseases of the myocardium in athletes, by discussing athlete-specific issues in each section of the manuscript, including epidemiology, pathophysiology, clinical signs and symptoms, diagnosis, and treatment. Furthermore, thanks to the reviewer’s suggestions, we have added some additional athlete-specific information in each section of the manuscript.

Point 2: The manuscript is too long and the Special Issue "Sudden Cardiac Death in Athletes" topic is not comprehensively addressed. The manuscript should be more focused on athletes and sudden cardiac death in the setting of myocarditis/perimyocarditis/inflammatory CMP.

Response 2: Thank you for your comment. Although we recognize that our manuscript is quite long, it now counts 8,375 words including tables, figures, legends, and bibliography, which is well under the 10,000 word limit usually reserved for review articles. You can find information on the risk of sudden death and ventricular arrhythmias in myocardial inflammatory diseases in the “epidemiology” (page 2, lines 59-63), “clinical signs and symptoms” (page 4, lines 156-170), and “comprehensive patient care” (page 10, lines 524-541, pages 10-11, lines 551-593) sections. In addition, we aimed at providing a general framework for the optimal treatment of athletes with myocardial inflammation, which is instrumental for the prevention of sudden cardiac death. Actually, our manuscript resulted lengthened after the revision process, since we provided more athlete-specific data and discussed mixed myocardial-pericardial syndromes and differential diagnoses; however, information is organized in several paragraphs, which should make the paper more readable.

Point 3: As we often face perimyocarditis I the acute face of diseases this should be mentioned and discussed.

Response 3: Thank you for raising this important point. We have now added information on the diagnosis (page 1, lines 35-38, page 6, lines 313-317, page 6, lines 337-339, page 7, lines 369-371, pages 7-8, lines 396-412), treatment (page 10, lines 515-521), and sports eligibility assessment (page 10, lines 553-555) of athletes with perimyocarditis-myopericarditis.

Point 4: More information on COVID-19 myocarditis in athletes is needed taking into account current pandemics.

Response 4: Thank you for your insightful comment. In the revised version of our manuscript, we have added information on the epidemiology of myocardial/pericardial inflammation in COVID-19 (page 2, lines 66-68), as well as on the assessment of sports eligibility among athletes suffering COVID-19 (page 11, lines 574-581).

Minor Remarks:

Point 1: Please discuss on specific etiological factors in special sporting activities – orienteering (exposure to borreliosis), boxing (droplet infection), a lot of contacts during global sporting events, sporting activities in endemic areas and etc.

Response 1: Thank you for your helpful comment. According to your suggestion, we have added a discussion on special etiological factors pertaining to specific sports activities (page 2, lines 81-88).

Point 2: Please discuss sensitivity and specificity of CMR to detect myocarditis.

Response 2: Thank you for your comment. We have now added quantitative data on the diagnostic accuracy of CMR for a diagnosis of myocarditis (page 7, lines 392-394).

Point 3: Please discuss NSAIDs in the setting of myocarditis; BB in athletes.

Response 3: Thank you for your helpful comment. We have added specific sections concerning the role of nonsteroidal anti-inflammatory drugs in mixed myocardial-pericardial inflammation as well as in myocarditis (page 10, lines 515-521), and beta-blockers in athletes (page 10, lines 511-514).

Point 4: Please stress that EMB should be performed in experienced centers…what is complication rate in athletes of EMB?

Response 4: Thank you for your insightful comment. We lack large-scale data concerning the safety of EMB in athletes, since most published series only included few subjects (see Dello Russo A, et al. Concealed cardiomyopathies in competitive athletes with ventricular arrhythmias and an apparently normal heart: role of cardiac electroanatomical mapping and biopsy. Heart Rhythm 2011, 8(12):1915-22 and Narducci ML, et al. Role of extensive diagnostic workup in young athletes and nonathletes with complex ventricular arrhythmias. Heart Rhythm 2020, 17(2):230-237). We have now stressed the concept that EMB, as well as the assessment of athletes with suspected myocardial inflammation, should be performed in experienced centers, due to the multiple challenges encountered in the management of these special patients (page 5, lines 237-238; page 5, lines 256-258; page 11, lines 603-605).

Point 5: Please include coronary artery CT angiography as a imaging test that could replace invasive angiography in athletes.

Response 5: Thank you for your comment. We have added a discussion on the potential role of coronary computed tomography angiography as a substitute for invasive coronary angiography in young athletes (page 9, lines 450-452).

Point 6: Please discuss more the timing of device implantation in acute myocarditis.

Response 6: Thank you for your helpful comment. You can now find a discussion on the optimal timing of cardiac implantable electronic device in patients with acute myocarditis at page 10, lines 528-530.

Point 7: Please talk about prevention measures in athletes (training activities during viral infections and etc., evidence of ICD shocks in athletes and etc.)

Response 7: Thank you for your insightful comment. We have added some data on the role of preventive measures in athletes, by stressing the concept that rest is recommended among athletes with COVID-19 (and other viral illnesses potentially associated to myocardial inflammation) even when the disease runs asymptomatically (page 11, lines 574-582), and on the importance of optimal ICD programming, in order to avoid inappropriate shocks (page 10, lines 535-537).

Round 2

Reviewer 2 Report

The manuscript "Myocardial inflammation, sports practice, and sudden cardiac death: 2021 update" has been significantly improved.